# UNDERSTANDING THE ROLE OF SELF ATTENTION FOR EFFICIENT SPEECH RECOGNITION

**Kyuhong Shim**[1], **Jungwook Choi**[2], **Wonyong Sung**[1]
Department of Electrical and Computer Engineering, Seoul National University[1]
Department of Electrical Engineering, Hanyang University[2]
`skhu20@snu.ac.kr, choij@hanyang.ac.kr, wysung@snu.ac.kr`

## ABSTRACT

Self-attention (SA) is a critical component of Transformer neural networks that have succeeded in automatic speech recognition (ASR). In this paper, we analyze the role of SA in Transformer-based ASR models for not only understanding the mechanism of improved recognition accuracy but also lowering the computational complexity. We reveal that SA performs two distinct roles: *phonetic* and *linguistic* localization. Especially, we show by experiments that phonetic localization in the lower layers extracts phonologically meaningful features from speech and reduces the phonetic variance in the utterance for proper linguistic localization in the upper layers. From this understanding, we discover that attention maps can be reused as long as their localization capability is preserved. To evaluate this idea, we implement the *layer-wise attention map reuse* on real GPU platforms and achieve up to 1.96 times speedup in inference and 33% savings in training time with noticeably improved ASR performance for the challenging benchmark on LibriSpeech dev/test-other dataset.

## 1 INTRODUCTION

Recent advances in end-to-end automatic speech recognition (ASR) have been driven by Transformer models (Vaswani et al., 2017). Transformer was first introduced for natural language processing (NLP) tasks such as neural machine translation (Vaswani et al., 2017; Ott et al., 2018), language modeling (Dai et al., 2019; Rae et al., 2019), and text generation (Raffel et al., 2020). Thanks to its superior performance in processing sequence input, Transformer has been widely adopted in various state-of-art ASR models (Zhang et al., 2020b; Ng et al., 2021; Guo et al., 2021). Self-attention (SA) is a core component of Transformer-based ASR, which dynamically collects information from multiple frames of an audio sequence. However, the computation and memory costs of SA increase quadratically with the length of a sequence, which is particularly problematic for ASR. For example, just a 30-second utterance corresponds to about 750 frames with a widely used window stride of 40ms.

Understanding the role of SA may provide essential insights for the efficient design of Transformer-based ASR models. Extensive studies have examined the behavior of SA in the field of NLP (Kovaleva et al., 2019; Park et al., 2019; Gong et al., 2019; Rogers et al., 2020). Recently, several studies further attempted to discover the characteristics of SA in the speech domain. Yang et al. (2020) revealed that self-attention features in the self-supervised audio Transformer are categorized into global, vertical, and diagonal patterns. Zhang et al. (2021b) focused on the diagonality of upper SA layers in ASR models for improving efficiency. However, these prior works revealed limited insights on the patterns discovered in SA, constraining its use for improving model efficiency. Thus, providing a holistic view of the role of SA is desirable for efficient ASR model design.

In this work, we reveal that SA plays two distinct roles in the success of Transformer-based ASR models: phonetic and linguistic localization, as illustrated in Figure 1. First, *phonetic localization* of lower SA layers attends to the phonologically meaningful global context. Second, *linguistic localization* of upper SA layers mainly attends to the local context of a near-diagonal attention map. We hypothesize that the phonetic variance in utterances such as variations in pronunciation is standardized in the lower SA layers so that the upper SA layers can identify local linguistic features

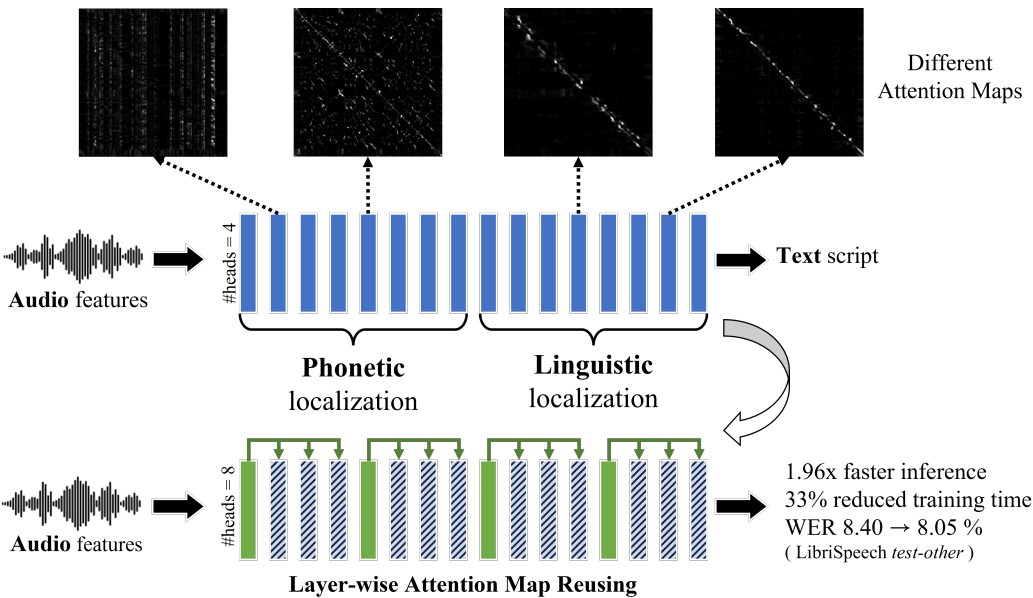

Figure 1: Illustration of the role of SA layers in Transformer-based ASR models and the proposed layer-wise attention map reuse. We discover that lower layers and upper layers show different behavior.

for accurate transcription. To investigate the behavior of SA layers, we propose *phoneme attention relationship* (PAR) to explain how phoneme localization works quantitatively. Interestingly, we discover that the phonetic localization represents the traditional knowledge on phonetics; for example, labial, velar, or nasal phonemes tend to attend to each other.

Based on this understanding, we propose a practical method for efficient ASR model design. We reuse the attention maps of SA layers while preserving the phonetic localization capability of the lower SA layers, resulting in up to 1.96x speedup in inference and 33% savings of training time with considerably improved performance in challenging ASR tasks (LibriSpeech dev/test-other dataset).

Our contributions can be summarized as follows:

- We reveal that SA layers contribute to ASR with two distinct roles: phonetic localization in the lower layers and linguistic localization in the upper layers. This unique distinction leads to an in-depth analysis of phonetic SA for the first time. We further propose phoneme attention relationship (PAR) to quantitatively identify the role of phonetic localization.

- We propose layer-wise attention map reuse for efficient Transformer-based ASR models. In particular, we discover that attention map reuse is possible in lower SA layers as long as the phonetic localization property quantified by PAR is preserved. We demonstrate with the popular ASR model and dataset that ASR performance can be maintained or slightly improved even if the attention map is reused.

- We implement the attention map reuse on real GPU platforms and achieve up to 1.96x inference speedup and 33% savings in training time, demonstrating that the proposed method is practical.

## 2 BACKGROUND

### 2.1 ASR ENCODER AND SELF-ATTENTION

SA is usually utilized as a module inside the ASR model consisting of stacked Transformer encoder layers. ASR encoder takes a sequence of short-time Fourier-transformed (STFT) audio features, known as a 'frame', as input and extracts a high-level feature of each frame through multiple layers. As illustrated in Figure 1, the extracted high-level feature changes for each layer; stacked SA layers

first extract phonetic features from audio features and utilize these features to build linguistic features for the output transcription.

We briefly review the SA[1] computation procedure. Consider a sequence of $d$-dimensional column vectors $X = \{x_1, x_2, ...x_T\}$ as input. Each vector corresponds to each frame of speech where the total number of $T$ frames are included. The input feature vector $X$ is projected to query $(Q)$, key $(K)$, and value $(V)$ of $h$-th attention head as follows:

$$q_{h,i} = W_h^Q x_i, \quad k_{h,i} = W_h^K x_i, \quad v_{h,i} = W_h^V x_i \quad \left(W_h^Q, W_h^K, W_h^V \in \mathbb{R}^{d_h \times d}\right) \quad (1)$$

$W^{Q,K,V}$ indicates projection matrices for $Q$, $K$, $V$, respectively. $d_h = d/H$ is the dimension of each attention head where $H$ is the number of attention heads. The attention map $(A_h)$, which represents how much frames attend to each other, is computed by scaled dot-product operation followed by softmax. The resulting attention map takes a form of a 2D matrix where each row is a probability vector. A single element of the attention map $(A_h[i,j])$ represents how much $i$-th frame attends to $j$-th frame. The attention head $(d_h)$ is a weighted sum of $V$ using the attention map as weight. Note that each attention head corresponds to a different attention map $A_h$; this multi-head design enables focusing on various perspectives within a single SA layer. The output $O = \{o_1, o_2, ...o_T\}$ is computed by the projection $(W^O \in \mathbb{R}^{d \times d})$ on the concatenated attention heads[2].

$$A_h[i,:] = \underset{j}{\text{Softmax}} \left(\frac{q_{h,i} k_{h,j}^T}{\sqrt{d_h}}\right), \quad d_{h,i} = \sum_{j=1}^{T} A_h[i,j] v_{h,j}, \quad o_i = W^O \underset{h}{\text{Concat}}(d_{h,i}) \quad (2)$$

SA layer contains $O(d^2)$ parameters. As shown in equations, SA requires quadratic computation and memory complexity $O(T^2)$, in exchange for the ability to access any location in the sequence. When $N$ layers of SA are stacked, the burden proportionally increases.

## 2.2 PREVIOUS WORK ON SELF-ATTENTION ANALYSIS

ASR considers both phonetic and linguistic aspects to transform audio input to text output. However, the studies on NLP mostly analyze the linguistic characteristics of SA, and the studies on self-supervised audio representation learning (SSAL) mainly focus on the phonetic behaviors of SA. The valuable findings from both domains cannot be directly applied to ASR.

**NLP** The behavior of SA has been widely studied in the NLP domain (Rogers et al., 2020), mostly focused on BERT (Devlin et al., 2019), a self-supervised language representation learning model. Kovaleva et al. (2019) and Guan et al. (2020) suggested that attention patterns can be clustered into several groups and the pattern may change depending on the fine-tuning task. Clark et al. (2019) and Tenney et al. (2019) observed attention maps that correspond to linguistic concepts of the language. Voita et al. (2019) also characterized linguistic attention heads and connected the knowledge to efficient model structure. However, studies on NLP only provide analysis on linguistic attention.

**SSAL** Recently, several studies have been introduced to understand how the audio information is encoded in SSAL models, such as CPC (Oord et al., 2018), Wav2Vec 2.0 (Baevski et al., 2020), Mockingjay (Liu et al., 2020), HuBERT (Hsu et al., 2021), and Audio ALBERT (Chi et al., 2021). Ma et al. (2021) and Shah et al. (2021) demonstrated that a wide spectrum of phonetic information is included in these models. Especially, Yang et al. (2020) categorized attention maps into three categories: global, vertical, and diagonal, where diagonal heads attend to local frames and vertical heads either focus or neglect specific phonemes. However, Yang et al. (2020) only discovered attention patterns without an explanation on how phonetic feature extraction is achieved with these patterns.

**ASR** Previous works have investigated the redundancy of attention maps mainly based on diagonality. From the observation that attention maps in upper layers show highly diagonal patterns, Zhang et al. (2021b) proposed replacing upper SA layers to feed-forward layers without performance loss. Zhang et al. (2021a) removed SA heads of high diagonality during training as a regularization

---

[1]In this paper, SA is equivalently used as a multi-head SA.
[2]The two terms 'head' and 'attention head' are used interchangeably.

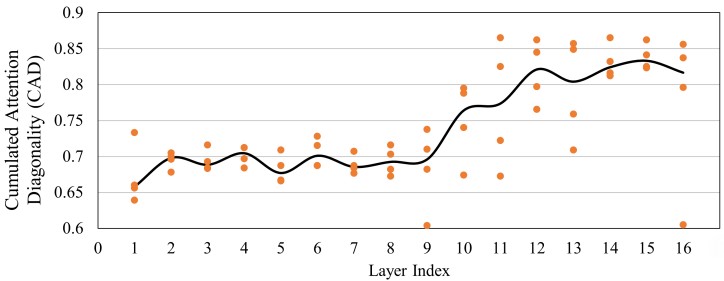

Figure 2: Cumulative attention diagonality (CAD) of each attention head. Four points for each layer correspond to the CAD of four attention heads. The black line connects the median across the layers.

but keep every head for the test time. Similarly, Chang et al. (2020) introduced adaptive attention span where each attention head equips a different attention span width to reduce the sequence length for the computation, starting from the intuition that some heads only attend to neighboring frames. These approaches mainly focus on reducing the burden of diagonal and concentrated attention, however, diagonality-based analysis has limitations in optimizing phonetic attention. We distinguish SA into two groups and provide proper analysis for each.

# 3 UNDERSTANDING THE ROLE OF SELF-ATTENTION IN ASR

## 3.1 ANALYSIS SETUP

We train and evaluate the model on the LibriSpeech-960 (Panayotov et al., 2015) dataset. The dataset include two types of data, *clean* and *other*, where *other* contains more challenging utterances. We extract the 80-dimensional log-Mel filterbank feature from a 25ms window with a stride of 10ms. We use 128 sub-word tokens as vocabulary, built on SentencePiece (Kudo & Richardson, 2018) library using the byte-pair encoding (Sennrich et al., 2016). The analyses are performed on LibriSpeech *test-clean* dataset unless specified.

We use Conformer-M(medium) (Gulati et al., 2020) as the baseline ASR encoder, trained with CTC (Graves et al., 2006) loss. Conformer is a variant of Transformer, which combines an additional convolution module to enhance the ability to collect local neighboring features. We follow recently introduced state-of-the-art ASR studies (Zhang et al., 2020b; Ng et al., 2021; Guo et al., 2021) that have adopted Conformer as their ASR encoder. We train the model with AdamW (Loshchilov & Hutter, 2018) optimizer for 200K iterations. Please see Appendix A.1 and A.2 for the model configuration and training details.

## 3.2 DISTINGUISHING BETWEEN PHONETIC AND LINGUISTIC SELF ATTENTIONS

To understand the role of SA, we start by examining the attention map $A_h$, which characterizes the functionality of SA. The attention map indicates how a frame attends the other frames in terms of probabilistic distribution for each attention head. Thus, analysis on attention maps across the heads of a SA layer would discover important characteristics of SA. To measure the diagonality of the attention map, we introduce *cumulative attention diagonality* (CAD) defined as the integral of the sum of attention probabilities constrained by distance as below:

$$\text{CAD}_h = \int_{r=0}^{1} \frac{1}{T} \sum_{i=1}^{T} \sum_{j=1}^{T} A_h[i,j] \cdot \mathbb{I}[|i-j| \leq r(T-1)] \mathrm{d}r \tag{3}$$

where $T$ is the number of frames in a sequence, $r$ determines the range of distribution in the attention map under test, and $h$ is an index of the attention head. Appendix B.1.1 provides detailed explanations of above equation and visualizes some CAD examples.

Figure 2 shows the attention diagonality analysis for the SA layers. There is a clear transition of CAD from lower layers (layer 1-8) to upper layers (layer 9-16). We emphasize this is a unique trend observed in ASR models compared to other domains: Kovaleva et al. (2019) classified attention maps in BERT but did not report the grouping of the same types. Yang et al. (2020) also categorized attention maps in SSAL but those categories broadly appear through layers. In ASR, Zhang et al. (2021b) observed a similar diagonality pattern for SA layers, but they considered it as an increase of diagonality over layers and did not separate the distinct patterns.

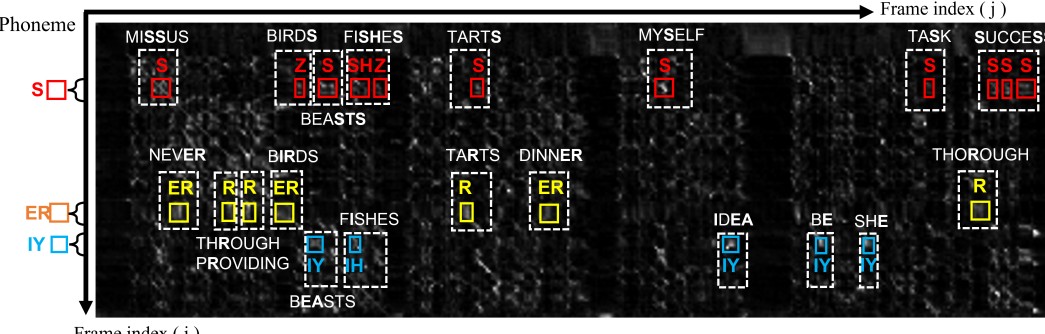

Figure 3: Visualization of the phonetic localization. Each element corresponds to $A[i, j]$ where $i, j$ indicates the frame index. Several rows that correspond to a certain phoneme, give higher attention to similar phonemes across the columns. For better visualization, we selectively draw boxes on three representative patterns (S, ER, and IY).

From the diagonality analysis, we categorize the role of SA layers into two parts: phonetic and linguistic localization. In ASR, linguistic localization refers to the behavior of the attention map that focuses on local (near in distance) frames and aggregates the information for text transcription. Since these local frames are particularly important in audio to text transcription in ASR, the attention map is characterized with a diagonal pattern. As shown in Figure 2, the upper layers tend to exhibit diagonally dominant patterns, implying their role as a linguistic localizer. We visualize two cases of diagonal attention patterns in Figure 1, extracted from layers 12 and 15 of the baseline model. Our observation is consistent with the prior work such as Zhang et al. (2021b). On the other hand, there has been little discussion about the role of lower layers in ASR in the context of phoneme localization. We discuss this in detail in the next section.

### 3.3 CHARACTERISTIC OF PHONETIC LOCALIZATION

Phonetic localization denotes the role of the attention map that focuses on similar (near in content) frames and extracts the phonologically meaningful features. We observe two characteristics of phonetic localization. First, phonetic localization is realized as attention to similar phonemes across the sequence. Second, the localization transforms each corresponding frame more likely to others.

To analyze phonetic attention, we exploit the phoneme information to find out the relationship between attention and phonemes. For the LibriSpeech dataset, we use the frame-level phoneme alignments obtained from Montreal Forced Aligner (McAuliffe et al., 2017). Table 1 lists all phoneme classes. Please visit Appendix A.3 for details on phoneme pre-processing.

| Idx. | 0 | 1 | 2 | 3 | 4 | 5 | 6 | 7 | 8 | 9 | 10 | 11 | 12 | 13 | 14 | 15 | 16 | 17 |
|------|----|----|----|----|----|----|----|----|----|----|----|----|----|----|----|----|----|----|
| Phn. | AA | AE | AW | AY | AH | EH | ER | EY | IY | IH | O | UH | UW | L | R | M | N | NG |
| Idx. | 18 | 19 | 20 | 21 | 22 | 23 | 24 | 25 | 26 | 27 | 28 | 29 | 30 | 31 | 32 | 33 | 34 | 35 |
| Phn. | B | D | DH | G | K | P | T | F | CH | SH | TH | S | Z | V | JH | W | Y | HH |

Table 1: Phoneme index for the analysis. Phonemes are extracted from the LibriSpeech lexicon and collapsed into 36 classes. Phonemes are reordered according to their phonological properties.

Figure 3 demonstrates the first characteristic of phoneme localization. The attention map presents that the same or similar phonemes tend to assign high attention weight to each other, for example, ('S' to 'Z'), ('ER' to 'R'), and ('IY' to 'IH').

We evaluate our second statement by phoneme classification on hidden layer representations (output of layers, also known as a hidden activation or hidden vector), similar to previous approaches (Baevski et al., 2021; Liu et al., 2021). We extract hidden layer representations and train a softmax classifier for each layer. Input is 256-dimensional vector and output contains total 37 output classes (36 phonemes + "silence"). See Appendix A.5 for the details on training the classifier and visualization of the confusion matrix.

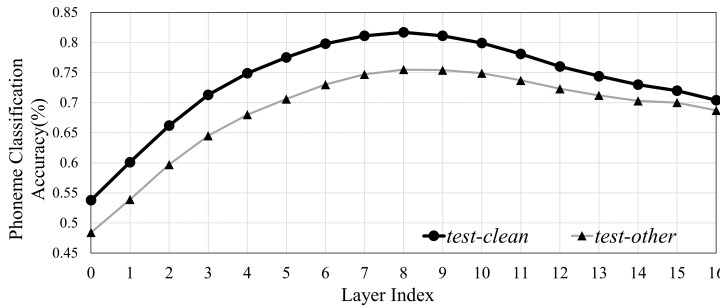

Figure 4: Phoneme classification accuracy on LibriSpeech *test* datasets. The zeroth entry implies the classifier is trained from hidden representations obtained before the first SA layer.

Figure 4 shows the phoneme classification accuracy for different layers. The accuracy increases for the lower layers (layer 1-8) where phonetic localization dominates. Specifically, the accuracy of layer 0 (before SA) is only 53.8%, but it consistently increases to 81.7% at layer 8. The phoneme classification accuracy indicates how well hidden vectors can be distinguished according to their phoneme classes. Therefore, the accuracy increase on lower layers implies that hidden layer representations are more standardized by phonetic localization. We assume that the accuracy decreases for the upper layers because they perform the linguistic localization and convert phoneme-level information to output text.

## 3.4 LAYER-WISE ANALYSIS OF PHONETIC ATTENTIONS

In phonetic attention, we observed that similar phonemes attend to each other and the phonetic features are clustered through layers. For the next step, we investigate the layer-wise behavior of phonetic attention to understand the contribution of each phonetic attention layer. We introduce *phoneme attention relationship* (PAR) to understand how SA processes phonological information by exposing how much each phoneme class attends to the other phonemes on average. Specifically, we directly map an attention probability $A_h[i, j]$ to $P_h[p, q]$ where $i$-th frame and $j$-th frame correspond to phoneme $p$ and $q$, respectively. If two phoneme classes $p$ and $q$ are different, we simply transport the probability from $A_h[i, j]$ to $P_h[p, q]$. On the other hand, if two frames are within the same phoneme class $p$, we exclude consecutive frames of the same class to avoid unnecessarily emphasize the effect of diagonal attention maps. PAR is computed as below[3]:

$$P_h[p, q] = \frac{T}{|C_p| \cdot |C_q|} \sum_{i \in C_p} \sum_{j \in C_q} A_h[i, j] \quad (p \neq q) \tag{4}$$

$$P_h[p, p] = \frac{T}{|C_p|} \sum_{i \in C_p} \frac{1}{|C_p| - |E_p(i)|} \sum_{j \in C_p - E_p(i)} A_h[i, j] \quad (p = q) \tag{5}$$

$C_p$, $C_q$ indicates the set of frame indices that correspond to phoneme class $p$ and $q$. $E_p(i)$ indicates the number of frames that satisfies two conditions: belong to the same class $p$ as $i$-th frame and all frames between itself and $i$-th frame also belong to the same class. In other words, $C_p - E_p(i)$ indicates the subset of $C_p$ that are not connected to $i$-th frame by consecutive class $p$ frames. For the $P_h[p, p]$ calculation, we do not include consecutive frames of the same class. We discuss the purpose of this exclusion in Appendix B.2.1.

We visualize the average PAR of the lower half and upper half of layers in Figure 5. A prominent diagonal component appears, representing that phonemes put high attention to themselves. Interestingly, we also discover well-known phonological characteristics in the lower half of layers. For example, labial (B, P), velar (G, K), and alveolar (S, Z) consonants highly attend to each other. Nasal phonemes (M, N, NG) also show a high correlation. From the empirical observations, we denote that phonetic attention map creates heterogeneous patterns. Thanks to the multi-head structure, a single SA layer can capture multiple phonetic relationships. Please see Figure 12 in Appendix B.2.2 for various PAR examples for each head.

If each phonetic attention map corresponds to different relationships, can we reuse phonetic attention maps across multiple layers? We answer this question by introducing the PAR coverage, which

---

[3]PAR as well as CAD are averaged through the corpora.

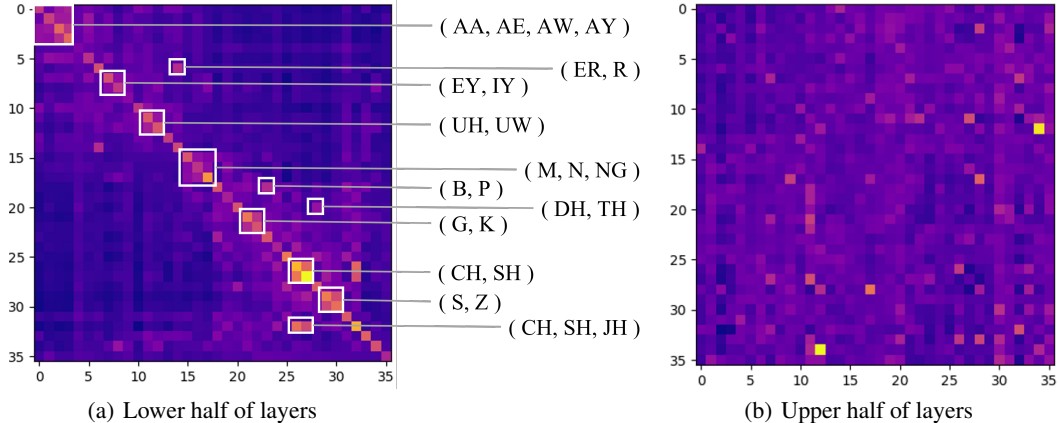

(a) Lower half of layers       (b) Upper half of layers

Figure 5: Averaged phoneme attention relationship of the lower half (1st, ... 8th) and upper half (9th, ... 16th) layers. The result is averaged through layers and heads on *test-clean* dataset. Each row and column corresponds to the phoneme index. Brighter (yellow) values indicate stronger attention between phonemes. Elements that stand out are highlighted, where phonemes with similar properties tend to attend to each other.

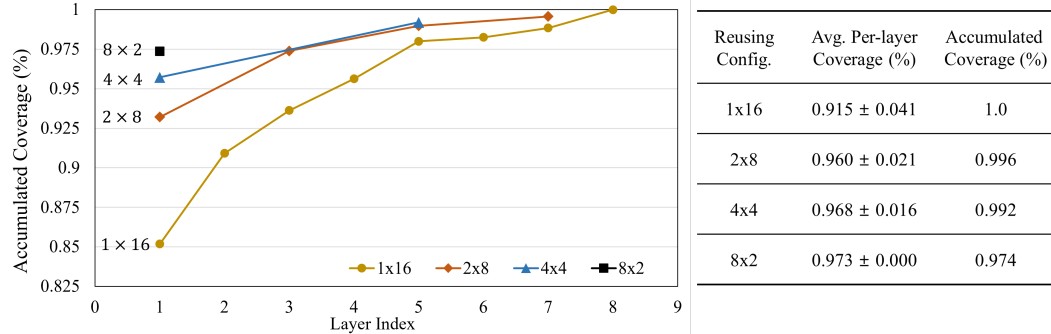

| Reusing Config. | Avg. Per-layer Coverage (%) | Accumulated Coverage (%) |
|---|---|---|
| 1x16 | 0.915 ± 0.041 | 1.0 |
| 2x8 | 0.960 ± 0.021 | 0.996 |
| 4x4 | 0.968 ± 0.016 | 0.992 |
| 8x2 | 0.973 ± 0.000 | 0.974 |

Figure 6: Accumulated PAR coverage of lower layers (1st, ... 8th). Averaged PAR of the baseline (Figure 5(a)) is set to 1.0, which is considered to be a desirable reference. The left plot on the accumulated coverage shows how each layer participates in covering the strength of the relationship. The right table summarizes the coverage of different reuse configurations. A higher average per-layer coverage ratio implies that each layer performs more similarly to the baseline.

indicates how much each layer covers the phonetic relationship represented in the averaged PAR from the baseline (Figure 5(a)). The averaged PAR of the baseline is considered to represent all the essential phoneme relationships. Please refer to Appendix A.4 for details.

To investigate the effect of reuse, we plot the accumulated PAR coverage for different reuse configurations in Figure 6. In calculating the accumulated coverage ratio, we take the average on every PAR under a certain layer and compare it with the reference PAR. The accumulated coverage consistently increases to 1, which means that the missing relationships are fulfilled through layers. We test four configurations (will be introduced in detail in the next Section) $X \times Y$, where $X$ layers share the same attention map. As the number of reuse increases ($2 \times 8 \rightarrow 4 \times 4 \rightarrow 8 \times 2$), the average per-layer coverage also grows, which implies that each SA layer tries to capture more phonetic relationships to recover the performance. However, for $8 \times 2$, the model fails to fully cover the reference, represented as a low accumulated coverage ratio of $0.974$. In Appendix B.3, we visualize the effect of the reuse on PAR coverage where phonetic features captured in $2 \times 8$ and $4 \times 4$ are missed in $8 \times 2$.

Table 2: Word error rate (%) for different attention map reuse configurations. "`HX`" indicates that the number of attention heads in the self-attention layer is set to `X`. All configurations carry almost the same number of parameters. No external language model is used.

| Configuration | #Heads | Head dim. | *dev-clean* | *dev-other* | *test-clean* | *test-other* |
|---|---|---|---|---|---|---|
| 1(`H4`) × 16 (baseline) | 64 | 64 | 3.1 | 8.3 | 3.2 | 8.4 |
| 2(`H4`) × 8 | 32 | 64 | **3.0** | 8.2 | 3.3 | 8.2 |
| 4(`H8`) + 4(`H8`) + 4(`H8`) + 4(`H8`) | 32 | 32 | 3.1 | **8.1** | **3.2** | **8.1** |
| 4(`H2`) + 4(`H2`) + 4(`H4`) + 4(`H4`) | 12 | 128/64 | 3.1 | 8.2 | 3.4 | 8.3 |
| 4(`H4`) + 4(`H4`) + 4(`H4`) + 4(`H4`) | 16 | 64 | **3.0** | 8.2 | 3.3 | 8.2 |
| 4(`H8`) + 4(`H8`) + 4(`H4`) + 4(`H4`) | 24 | 32/64 | 3.1 | 8.3 | **3.2** | 8.4 |
| 4(`H4`) + 4(`H4`) + 4(`H2`) + 4(`H2`) | 12 | 64/128 | 3.1 | 8.5 | 3.4 | 8.5 |
| 4(`H4`) + 4(`H4`) + 4(`H4`) + 4(`H4`) | 16 | 64 | **3.0** | 8.2 | 3.3 | 8.2 |
| 4(`H4`) + 4(`H4`) + 4(`H8`) + 4(`H8`) | 24 | 64/32 | 3.1 | 8.2 | 3.3 | 8.1 |
| 4(`H4`) + 4(`H4`) + 8(`H4`) | 12 | 64 | 3.1 | 8.3 | 3.3 | 8.2 |
| 8(`H4`) + 4(`H4`) + 4(`H4`) | 12 | 64 | 3.1 | 8.5 | 3.3 | 8.6 |
| 8(`H4`) + 8(`H4`) | 8 | 64 | 3.3 | 8.8 | 3.6 | 8.7 |
| 8(`H8`) + 8(`H8`) | 16 | 32 | 3.2 | 8.5 | 3.4 | 8.5 |

## 4 LAYER-WISE ATTENTION MAP REUSE

We propose *layer-wise attention map reuse*, a method to design an efficient Transformer-based ASR encoder by reducing the heavy SA computation. The core idea is to reuse the computed attention map from the previous layer. More specifically, we reuse attention map of $l$-th SA layer to $(l+1), (l+2), ...(l+M-1)$-th consecutive SA layers. If a single attention map is shared through $M$ layers, the computation burden of SA can be reduced by $M$ times. During training, the reused attention map receives gradients from $M$ layers. This layer-wise reuse is easy to implement and fully supported by modern accelerator hardware. The idea of reuse attention map through layers have been proposed for NLP (Xiao et al., 2019; Ying et al., 2021) but not tested for ASR. We discuss the difference in Section 5. For SA layers that receive the pre-computed attention map, query and key are not used and can be removed. To compensate the parameter size for those layers, we simply double the output dimension of $V$ ($W_h^V \in \mathbb{R}^{2d_h \times d}, W^O \in \mathbb{R}^{d \times 2d}$).

Table 2 shows the word error rate (WER) on LibriSpeech *dev* and *test* dataset. Configuration "$X \times Y$" indicates that $X$ successive layers are grouped to share the same attention map and total $Y$ groups are built. Therefore, there exist $HXY$ unique attention heads for each model. We also use the notation '+', for example, $4 \times 4$ is identical to $4 + 4 + 4 + 4$. For each configuration, we train the model from scratch with the same training setup as the baseline. Note that the increased number of heads comes with the decreased per-head dimension to keep the parameter size comparable.

**Best and Worst** We first compare the best (4(`H8`) × 4) and the worst configuration (8(`H4`) × 2). The worst is the most naive setting that just applies very aggressive attention map reuse. Although the speed is about the same, performance can be improved by increasing the number of heads (8(`H8`) × 2). In contrast, the best working setting is carefully designed to maximize performance. For example, equipping the same number of heads (2(`H4`) × 8) does not show similar performance compared to the best configuration.

**Sensitivity to Reuse** We examine which of the phonetic or linguistic localization is more sensitive to attention map reuse. Comparing two configurations with the identical number of heads and head dimensions ($4+4+8$ vs. $8+4+4$) in the 5th block of Table 2, we conclude the phonetic localization suffers more from increasing the reuse of layers. In other words, the linguistic localization seems to be more robust to the reuse. We conjecture that too few phonetic localization heads fail to capture every essential relationship.

**Number of Heads in Phonetic Localization** To better understand the trade-off between the number of heads and head dimension, we conduct three experiments that only differ on the number of heads in lower layers. As shown in the 3rd block of Table 2, among the three configurations,

Table 3: Effect of different configurations on speed. The numbers inside of the parentheses indicate the speed-up ratio. The front convolutional sub-sampling is not included. Changing the number of heads does not make much difference to the speed.

| Config. | #Param (M) | Latency(ms) by sequence length (approx. sec) | | | | Training cost(h) |
|---|---|---|---|---|---|---|
| | | 128 (5.1s) | 256 (10.2s) | 512 (20.5s) | 768 (30.7s) | |
| $1 \times 16$ | 25.45 | 1.43 (x1.00) | 3.74 (x1.00) | 11.11 (x1.00) | 22.32 (x1.00) | 430.0 |
| $2 \times 8$ | 24.92 | 1.25 (x1.15) | 2.98 (x1.26) | 7.92 (x1.40) | 15.05 (x1.48) | 337.5 |
| $4 \times 4$ | 24.66 | 1.14 (x1.25) | 2.56 (x1.46) | 6.29 (x1.77) | 11.38 (x1.96) | 288.4 |
| $8 \times 2$ | 24.52 | 1.08 (x1.32) | 2.35 (x1.59) | 5.47 (x2.03) | 9.55 (x2.34) | 268.8 |

$(4(\texttt{H4}) \times 4)$ surpasses the other two in 3 over 4 benchmarks. We expect a trade-off between the number of phonetic localization heads and per-head dimension; the former enables more various aspects to be covered while the latter helps richer representation for each head.

**Number of Heads in Linguistic Localization** We perform the same experiment as above for the upper layers, shown in the 4th block of Table 2. Interestingly, we found that increasing the number of heads for linguistic localization tends to improve the overall performance. In addition, a considerable performance loss is detected when the number of heads is decreased to $2(\texttt{H2})$. This observation implies that the previous studies that only reduce linguistic attention may face limitations when the remaining linguistic heads are too few.

**Inference and Training Speed** Table 3 compares different reuse configurations. Both training and inference speed can be greatly improved by reducing the number of attention computation. The impact becomes more significant for longer sequences as $T$ increases. Our best configuration $(4(\texttt{H8}) \times 4)$ accelerates the inference by 1.96x times (for 30-second utterance) and reduces training cost by 33%. Note that the number of parameters is almost equivalent for all configurations because of the expansion of $V$ dimension. Inference speed is evaluated on a single RTX-Titan(24GB) GPU and training cost is measured in GPU-hours on A100(40GB) GPU.

## 5 RELATED WORK

**Attention Map Reuse in NLP** Xiao et al. (2019) proposed sharing of attention map through consecutive layers for neural machine translation. They determine the reuse policy by Jensen-Shannon divergence (JSD) values between two attention maps. Ying et al. (2021) propose a similar approach for BERT but with manual reuse configurations. In addition to the critical difference in the domain (NLP vs. ASR), both works depend on the similarities of the attention map, however, do not investigate why the similarity is developed.

**Efficient Attention Map Computation for ASR** Several studies have been proposed to reduce the cost of attention computation for ASR. Wang et al. (2021) proposed a prob-sparse SA that only computes the top-k queries that are less uniform. For the streaming purpose, block processing of input sequence has been widely adopted (Yeh et al., 2021; Shi et al., 2021). Masked attention, which restricts the attention range to local neighbors, have also been used (Zhang et al., 2020a; Tripathi et al., 2020; Audhkhasi et al., 2021). While these approaches focus on reducing the effective sequence length, our method directly reduces SA computation.

## 6 CONCLUSION

In this paper, we analyze the role of self-attention in Transformer-based ASR and show that the role can be distinguished into two types: phonetic and linguistic localization. Especially, we showed that the phonetic localization captures various phonetic relationships and contributes to the performance by standardizing the features over similar phonemes, verified by the increasing phoneme classification accuracy over lower layers. The distinguished roles of SA in lower and upper layers also lead to an efficient ASR model that reuses the attention map for multiple SA layers. The proposed method has achieved a significant 1.96 times of speedup in inference and 33% reduced training time, with the reduction of word error rate from 8.40% to 8.05% on the LibriSpeech *test-other* dataset.

REPRODUCIBILITY STATEMENT

We explain extra details for the model architecture, training procedure, pre-processing steps, and experiments for analyses in Appendix A. We also provide the source code for the experiments in supplemental materials.

ACKNOWLEDGMENTS

This work was supported by the National Research Foundation of Korea (NRF) grant funded by Korea government (MSIT) (No. 2021R1A2C1013513). This work was also partly supported by Institute of Information & communications Technology Planning & Evaluation (IITP) grant funded by MSIT (No. 2020-0-01373, No. 2021-0-00020-001). This work was also supported in part by Samsung Advanced Institute of Technology, Samsung Electronics Co., Ltd. This work was also partly supported by the Google AI Focused Research Awards Program awarded to Wonyong Sung. We gratefully acknowledge the GCP credit support from Google AI and the GPU server support from the Artificial Intelligence Cluster Agency (AICA, Korea).

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

# A   DETAILS

## A.1   MODEL

Our baseline model is Conformer-M(medium) (Gulati et al., 2020) trained with CTC (Graves et al., 2006) loss. Table 4 shows the configuration of the model. Unspecified details follow the original Conformer paper. We observed that SyncBN (Peng et al., 2018) is critical for the overall performance. We employ weak attention suppression (WAS) (Shi et al., 2020) of $\gamma = 0.5$ for faster convergence and improved performance.

| Encoder | | | |
|---|---|---|---|
| #Layers | 16 | Hidden dim. | 256 |
| #Heads | 4 | Feed-forward dim. | 1024 |
| Conv. kernel size | 31 | Conv. normalization | SyncBN |
| BN momentum | 0.005 | BN epsilon | 1e-5 |
| Hidden drop prob. | 0.1 | Attention drop prob. | 0.1 |
| **Conv. Subsampling** | | | |
| #Layers | 2 | #Channels | 256 |
| Conv. kernel size | 3 | Conv. normalization | SyncBN |
| Conv. stride | 2 | Activation func. | ReLU |

Table 4: Conformer-M implementation details.

We decide to use Conformer as our baseline, following recent state-of-the-art ASR models, because these models can benefit the most from the proposed efficient model design. However, there may be several concerns on the clarity of our analysis on SA because the convolution module is jointly used with SA inside Conformer. Because the ability to gather information from the entire sequence is only equipped in SA, the analysis results on the role of phonetic heads could not be presented without SA. The convolution kernel size of 31, which covers about 1.2 seconds, is too short to gather long-range information. We believe that the convolution module may guide the model to focus on local information first at the early stage of the training, however, the role of SA is not much affected by the difference between Conformer and Transformer.

## A.2   TRAINING

| Optimizer & Scheduler | | | |
|---|---|---|---|
| Maximum LR | 1.5e-3 | Weight decay | 1e-5 |
| Adam epsilon | 1e-8 | Adam betas | (0.9, 0.99) |
| LR warm-up iters | 5K | LR keep iters | 95K |
| Total iters | 200K | Batch size | 480 |
| **Additional Details** | | | |
| #Frequency masking | 2 | Frequency mask width | 27 |
| #Time masking | 10 | Time mask width | 0.05 (5%) |
| #Models for SWA | 45 | Variational noise | 0.02 |
| CTC beam size | 32 | Gradient norm clip | 20 |

Table 5: Training details including optimizer, scheduler, augmentation and other hyper-parameters.

We use AdamW (Loshchilov & Hutter, 2018) optimizer with the inverse square-root learning rate schedule (Vaswani et al., 2017). Table 4 shows the training configuration. We linearly increase the learning rate (LR) to the maximum value for 5K iterations and keep LR at maximum for 95K iterations, followed by 100K iterations of LR decrease. We use 4x A100(40GB) GPUs for the experiments. To fit the batch size of 480 in these GPUs, we assign 40 samples per GPU and accumulate the gradient of 3 batches. We don't use bucketing for generating the mini-batch during training. We also employ adaptive SpecAugment (Park et al., 2020), stochastic weight averaging (SWA) (Izmailov et al., 2018), and variational noise.

A.3    PHONEME PRE-PROCESSING

For phonetic analyses, We employ the collapsed list of phonemes that are included in the LibriSpeech lexicon. We collapse ('AA, AO' to 'AA'), ('OW, OY' to 'O'), and ('SH, ZH' to 'SH'), which leads to the phoneme classes in Table 1, for better understanding the characteristics. Because the LibriSpeech dataset does not provide frame-wise phoneme alignments, we extract the phoneme alignment from MFA (McAuliffe et al., 2017) and map these alignments to each frame. Especially, we exploit the fact that each frame corresponds to a 40ms interval after passing through two convolutional layers (convolutional sub-sampling) of stride 2 in front of the model. We assign the phoneme class to each frame if the center of the frame is within the phoneme duration, including the 'silence' phoneme.

Except for the phoneme classification, our analysis excludes 'silence' frames by exploiting frame-level phoneme alignments. Then, we re-normalize the remaining attention probability to preserve the probability sum to 1. We observe that these silence frames sometimes consume too much probability mass for both linguistic and phonetic heads, which makes our analysis difficult. Note that this phenomenon of assigning strong attention to ambiguous tokens, such as [CLS] or [SEP], has been also reported in NLP (Kobayashi et al., 2020; Sun & Marasović, 2021). Recently, Kobayashi et al. (2020) introduced the concept of norm-based analysis and reported that those [CLS] and [SEP] does not contribute much to the output even though their attention weight is large. We leave analysis using effective attention as a future work.

A.4    PHONEME ATTENTION RELATIONSHIP COVERAGE

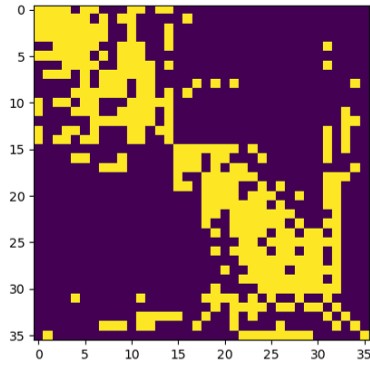

Figure 7: Top-10 phoneme classes (column) for each phoneme (row) in the reference phoneme attention relationship of Figure 5(a).

The coverage ratio $R$ of the target PAR compared to the reference PAR is calculated as below:

$$R_h[p, q] = \text{Minimum}\left(\frac{\text{PAR}_h[p, q]}{\text{PAR}_h^{\text{ref}}[p, q]}, 1\right), \quad R_h = \frac{1}{|P|}\sum_{p \in P}\frac{1}{|Q_p|}\sum_{q \in Q_p} R_h[p, q] \quad (6)$$

$P$ indicates every phoneme class and $Q_p$ indicates top-10 phoneme classes in the order of the largest PAR elements for the phoneme $p$ ($R_h[p, :]$). Top-10 classes are visualized in Figure 7. We exploit topmost phoneme classes because our interest is at the important relationships.

A.5    PHONEME CLASSIFICATION

For the phoneme classification, we train a single fully-connected layer as a classifier. We choose the simplest architecture as a classifier to more directly correlate the phoneme accuracy and hidden layer representations. These representations are extracted from *dev-clean* and *dev-other* dataset and evaluated on *test-clean* and *test-other* dataset. We use SGD with a learning rate of 0.1, momentum of 0.9, and weight decay of 1e-3. The training takes 15 epochs, where the learning rate is multiplied by 0.1 for every 3 epochs. We visualize confusion matrices of the phoneme classification in Figure 8. As layer proceeds, wrongly classified phonemes (non-diagonal) disappear and leave a clear diagonal line on the confusion matrix.

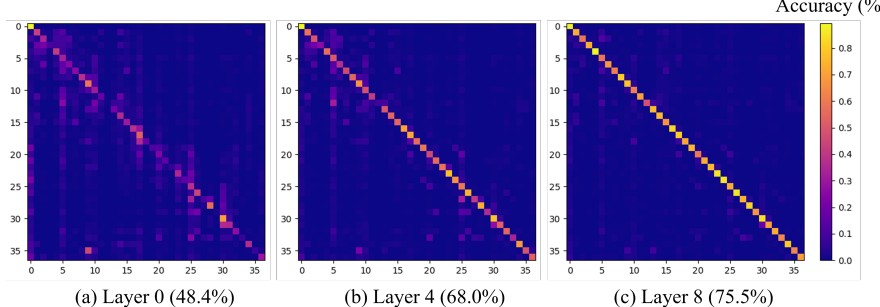

(a) Layer 0 (48.4%)     (b) Layer 4 (68.0%)     (c) Layer 8 (75.5%)

Figure 8: Confusion matrix and phoneme accuracy for selected layers. Visualized the result from the LibriSpeech *test-other* dataset. Each row and column corresponds to the 37 phoneme classes, including 'silence' as zeroth class.

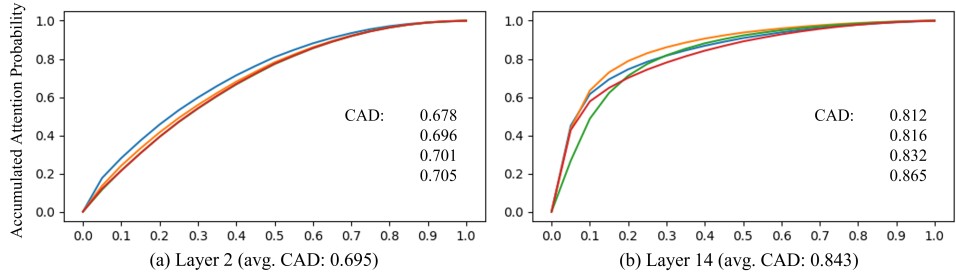

(a) Layer 2 (avg. CAD: 0.695)     (b) Layer 14 (avg. CAD: 0.843)

Figure 9: Examples of typical cumulative attention diagonality (CAD). (a) and (b) visualizes curves (before the integral) of four attention heads in each layer. $x$-axis and $y$-axis depict the relative distance $r$ and accumulated attention probability, respectively. CAD is represented as the area under curve. CAD values of each head are also listed.

## B ADDITIONAL RESULTS AND DISCUSSIONS

### B.1 DIAGONALITY

#### B.1.1 CAD RESULTS

The CAD is a good indicator of how fast the accumulated attention increases over the distance, directly represents the diagonality of the attention weight. The CAD is interpreted as the area under the function $D(r)$, where $D(r)$ calculates the amount of total attention weight within the restricted range $j \in [i - r(T-1), i + r(T-1)]$. $(T-1)$ is the maximum possible distance between two frames where $T$ is the number of frames.

$$\text{CAD}_h = \int_{r=0}^{1} \frac{1}{T} \sum_{i=1}^{T} \left( \sum_{\substack{j=\max(1, \\ i-r(T-1))}}^{\substack{\min(T, \\ i+r(T-1))}} A_h[i,j] \right) \mathrm{d}r = \int_{r=0}^{1} D(r)\mathrm{d}r \tag{7}$$

If the $r$ is same, a larger $D(r)$ means that the attention is more concentrated near the diagonal. Please note that $D(r)$ is a monotonically increasing function whose output is always in the range $[0, 1]$.

To help understand the concept of CAD, we visualize two typical examples of the cumulative attention diagonality in Figure 9. For layer 2, where attention heads perform the phonetic localization, CAD values are low. In contrast, for layer 14 that concentrates on linguistic localization, CAD values are much higher.

Figure 10 plots the sorted CAD values. We determined the threshold (0.75) where the curve of sorted CAD values changes from convex to concave. The higher CAD value represents that more probability mass is concentrated near the diagonal in the attention map, while the lower CAD value

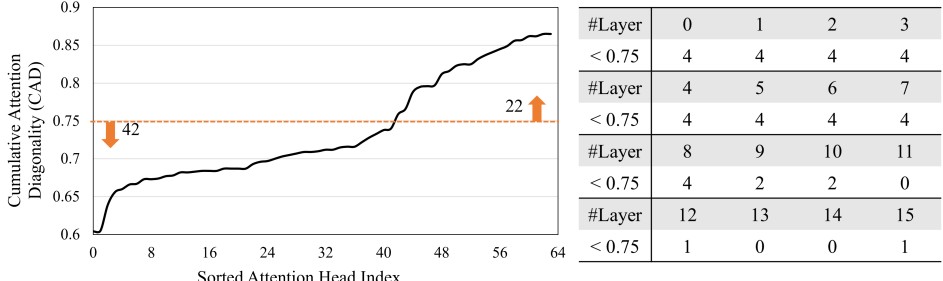

Figure 10: Sorted CAD over attention heads in the baseline model. Over 64 heads, 42 heads belong to the CAD value under $0.75$. The table on the right side indicates the number of linguistic attention heads that are of CAD value under $0.75$.

implies that the distribution is more uniformly distributed. We observe that diagonally concentrated heads take a large portion in upper layers.

### B.1.2 Previous Diagonality Analysis

Previous studies utilized a different metric to calculate the diagonality of an attention map (Zhang et al., 2021b; Yang et al., 2020). In this version of diagonality, the metric is interpreted as the negative normalized average attention distance (span-length). In Yang et al. (2020), the diagonality is calculated as below:

$$D_h = 1 - \frac{1}{T^2} \sum_{i=1}^{T} \sum_{j=1}^{T} A_h[i,j] \cdot |i-j| \tag{8}$$

We introduced the cumulative attention diagonality (CAD) because the previous diagonality metric lacks information about how attention is distributed by distance. In other words, CAD is more comprehensive because it provides the overall diagonality value as well as the tendency of the attention according to the distance. Our CAD metric dearly captures the flat region in lower layers (Figure 2). In contrast, in Zhang et al. (2021b), the diagonality increases from the lower layers to the upper layers, which may not be sufficient observation in ASR.

### B.2 Phoneme Attention Relationship

### B.2.1 Excluding Consecutive Frames

We treat $P_h[p,q](p \neq q)$ and $P_h[p,p]$ differently because we want to separate the effect of the diagonal (position-based) attention map. An attention map that highly focuses on the surroundings (diagonal-like) can unintentionally disturb the purpose of PAR in measuring $P_h[p,p]$, because it will give a high value to $p$-$p$ relationship not because the contents are similar, but because the location is close. Our purpose on PAR is to examine the phonetic (content-based) behavior, so excluding consecutive frames of the same phoneme class better represents SA in lower layers that correspond to the phonetic localization.

### B.2.2 PAR Results

Figure 11 visualizes how averaged PAR changes through layers. Lower layers (first row in Figure) show noticeable regions that represent the important phoneme relationships, including the diagonal. Note that the diagonal stands out even though we excluded the consecutive same phoneme classes. Each lower layer focuses on different aspects, supporting our analysis on the PAR coverage (Figure 6) that the accumulated coverage ratio continuously increases through layers. In other words, each layer covers a certain part of the reference PAR with less overlap between layers. On the other hand, the upper layers (second row in Figure) do not show the emphasized pattern. This is expected for linguistic localization heads that generate highly diagonal attention maps because their attention is mainly assigned to near frames regardless of their phoneme classes.

For the finer understanding, we also visualize how attention heads compose the averaged PAR. Figure 12 shows that each head corresponds to different phonological properties but their averaged

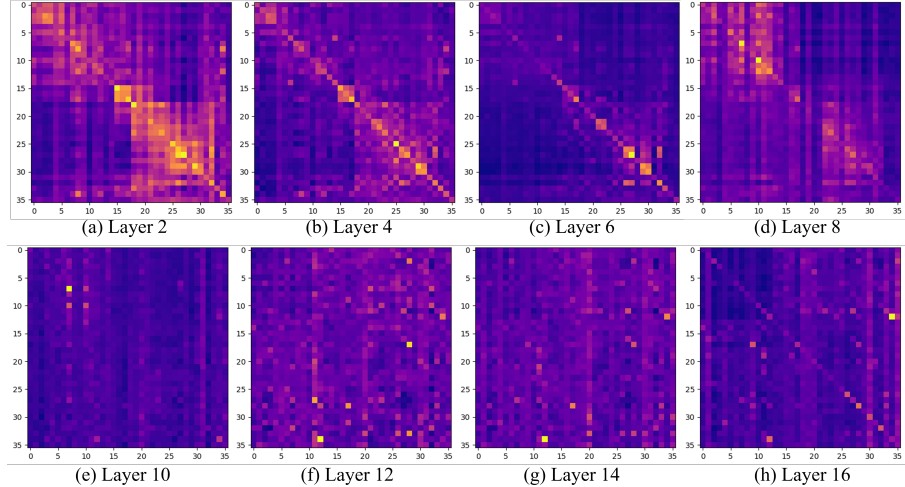

Figure 11: Phoneme attention relationships (PAR) in SA layers. Averaged PAR of even-numbered layers (2nd, 4th, ... 16th) are visualized. Lower layers show high correlations between similar phonetic features, but these relationships are weakened in upper layers.

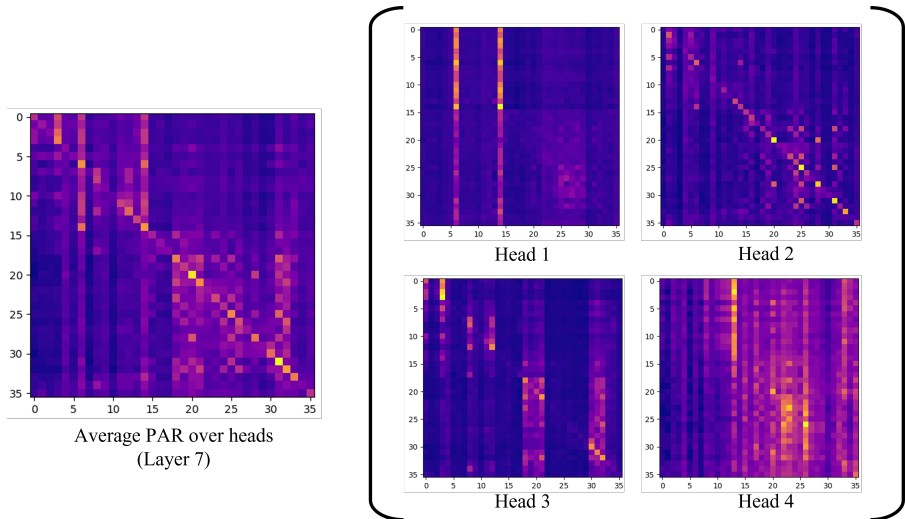

Figure 12: Phoneme attention relationships of self-attention heads. Each head see different aspects, enriching the overall information the layer captures.

interests mimic the reference PAR. This emphasizes the importance of multiple attention heads in phonetic localization; each head specializes in capturing the specific phoneme relationship and contributes differently.

### B.2.3 PREVIOUS PHONEME RELATIONSHIP ANALYSIS

The idea of phoneme relationship analysis is first introduced in Yang et al. (2020) as phoneme relation map (PRM), but our PAR is different from PRM in two ways. First and the most difference is that PRM do not distinguish consecutive frames and discontinuous frames that corresponds to the same phoneme class. Therefore, attention heads that only focus on neighbors would also present heavy self-to-self phoneme relationship in PRM, which hinders the clarity of the analysis. Second, PRM do not apply correction according to the sequence length. We multiply $T$ to reduce the effect of sequence length, inspired by the fact that the expectation of (averaged) probability is potentially smaller for longer sequences.

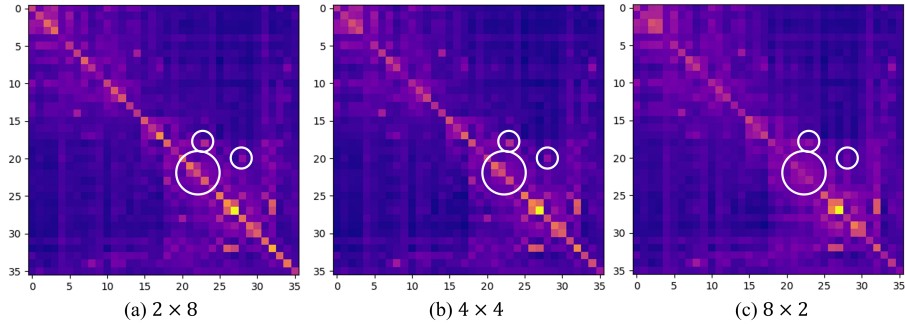

|                | (a) $2 \times 8$ | (b) $4 \times 4$ | (c) $8 \times 2$ |

Figure 13: Phoneme attention relationship (PAR) for three attention map reuse configurations ($2 \times 8$, $4 \times 4$, $8 \times 2$). PAR(a)(b)(c) are obtained by averaging PAR of lower layers. Key missing relationship is highlighted in white circles, such as (B, P), (G, K), and (DH, TH) in Figure 5(a).

## B.3   ATTENTION MAP REUSE AND PAR

For the understanding of the performance loss for extreme reuse cases, we visualize the PAR of different configurations in Figure 13. We observe that most of the patterns resemble the baseline model, which implies that heads learn similar roles during training. However, for $8 \times 2$, several information is lost; diagonal became unclear and highlighted correlations disappeared.

## B.4   COMPARISON TO MASKED ATTENTION

We compare the masked attention (Zhang et al., 2020a; Tripathi et al., 2020; Audhkhasi et al., 2021; Huang et al., 2020) with the proposed attention map reuse. Masked attention, similar to block-based attention (Shen et al., 2018; Qiu et al., 2020), is a method to reduce the computational burden of self-attention by restricting the length of the accessible context. If masked attention is adopted for ASR, each frame only attends to local neighbors; $L$ frames to the left and $R$ frames to the right, denoted as $[-L, R]$.

Table 6: Comparison of the word error rate between the proposed attention reuse and the masked attention. "-" indicates that the attention range is not restricted (unlimited).

| Model | Lower layers | Upper layers | *dev-clean* | *dev-other* | *test-clean* | *test-other* |
|---|---|---|---|---|---|---|
| Baseline | - | - | 3.1 | 8.3 | 3.2 | 8.4 |
| Low64 | $[-64, 64]$ | - | 3.1 | 8.4 | 3.4 | 8.5 |
| Up64 | - | $[-64, 64]$ | 3.1 | 8.2 | 3.3 | 8.2 |
| $4(\texttt{H8}) \times 4$ | - | - | 3.1 | 8.1 | 3.2 | 8.1 |

We train the models with the same setting as the Conformer-M baseline (as in Table 2) but with a limited attention range for either lower or upper layers. Among the 16 layers in the baseline, lower and upper layers consist of 8 layers each and correspond to phonetic localization and linguistic localization, respectively. For selected layers, we restrict each frame to only attend to near neighbors within the distance of 64 frames. Table 6 shows the performance on the LibriSpeech dataset.

For lower layers, attention range restriction causes a clear degradation of the recognition accuracy. We consider that phonetic localization in lower layers demands a wide range of attention. On the other hand, for upper layers, attention range restriction shows almost comparable performance to the baseline, and even better in some subsets. We hypothesize that the upper layers for CTC-based ASR may not require a very wide context, because their attention pattern is highly diagonal. In addition, the restriction-based computational savings seems to be no larger than the proposed attention reuse. A context range of 128 (64+64) frames corresponds to about 5.1 seconds, where the average utterance length of the corpora is about 7.4 seconds. Therefore, we expect approximately 30% reduction in attention calculation when masked attention is applied to every layer. Our attention reuse ($4 \times 4$) reduces about 75% of the attention computation without any degradation in the performance.

