# OpenReview forum: "Understanding the Role of Self Attention for Efficient Speech Recognition"
_ICLR.cc/2022/Conference — ICLR 2022 Spotlight_

### Official Review · Reviewer_mbFx · 2021-11-02

**Correctness:** 4
**Technical Novelty And Significance:** 3
**Empirical Novelty And Significance:** 3
**Recommendation:** 8
**Confidence:** 5

**Main Review:**

Strength

The analysis part of the paper can be a good contribution to the speech recognition community. The distinct increase after the first half of layers on cumulative attention diagonality brings an interesting insight on the behavior of the speech recognition model, and the characterization of phonetic and linguistic localization is convincing.

The phoneme analysis authors conduct in the paper provide a good insight for readers to understand the correlation between attention and phoneme classes.
The phoneme attention relationship is a good metric to understand the correlation between attention and phoneme, and provide a good measurement on the attention coverage for speech recognition

Overall the proposed analysis mechanism can benefit beyond improving inference speed, for example, it will be a great analysis tool for understanding representation learning which is beneficial for speech self-supervised learning.

Weakness

The downstream task this paper focuses on, improving speed with attention reuse, does not bring out the full potential of the proposed analysis mechanism. In fact, it seems the connection between the analysis and the task is not very strong, as one can design those configurations listed in Table 2 without the insight of phoneme attention relationship. It would serve authors better to identify other more correlated tasks.


**Summary Of The Paper:**

This paper develops a mechanism to analyze self-attention for speech recognition, and further use the resulting insight to facilitate attention reuse for reducing inference cost. The analysis identifies a distinct pattern between lower half and upper half layers on the ASR encoder through cumulative attention diagonality and characterizes the lower half as finding phoneme localization and upper half as finding linguistic localization. The paper justifies the characterization of phoneme localization by illustrating an example of such a phoneme localization, and also shows that the output of the layers in the middle provide better phoneme classification accuracy than the upper layers. An additional analysis, called phoneme attention relationship, further quantifies the correlation between phonemes and attention, and shows that the lower layers have higher correlation. These insights are used to design attention reuse across layers, and shown to maintain the quality while improving the speed on Librispeech.



**Summary Of The Review:**

The analysis proposed in this paper can be a good contribution to the community. The task focused in this paper, however, is on the weak side. Overall the analysis is beneficial to the community, and can be considered for acceptance.

---

> ### Author Response · Authors · 2021-11-16
> **Response to Reviewer mbFx**
>
> We appreciate the reviewer’s valuable comments.
>
> **Q**: The downstream task (ASR) does not bring out the full potential of the proposed analysis. One can design these (reusing) configurations without the insight of the PAR.
>
> Thank you for your insightful comments. After analyzing the attention pattern, we focused on attention map reuse because it can provide the most direct and tangible value to the speech community. Because our reusing approach starts from in-depth analysis, we could not only reduce the inference time but also find ways of improving the recognition accuracy further, such as increasing the number of heads (H4->H8) to cover more phonetic relationships in the lower layers.
>
> We can compare ours with other works [1][2][3]: all three find the reason for attention map reuse from the “high degree of similarity between different layers”. These works do not incorporate domain-specific analysis as ours, and therefore could not achieve the performance like ours: [1] targets NMT and achieved about x1.3 speedup with almost no decrease in BLEU score. [2] targets BERT and achieved about x1.3 speedup with slightly improved GLUE performance but without sufficient explanation. [3] targets various models (BERT, T5, ViT) but with a marginal speedup under 20%. Our method not only speeds up the inference (x1.96 times) significantly but also reduces WER on difficult datasets.
>
> We agree that our analysis, especially phonetic localization, can be a great tool to understand why Transformer-based models can achieve excellent performance. For example, it would be interesting to analyze how much the PAR (from Transformer) contributes to the recognition accuracy, compared to GMM (Gaussian Mixture Model), DNN, and RNN-based acoustic models. We consider that the ability to capture the long-range phoneme relationship would be critical. Motivated by your comments, we plan to conduct this experiment as soon as possible.
>
> *References:*
>
> [1] Xiao, Tong, et al. "Sharing attention weights for fast transformer." arXiv preprint arXiv:1906.11024 (2019).
>
> [2] Ying, Chengxuan, et al. "LazyFormer: Self Attention with Lazy Update." arXiv preprint arXiv:2102.12702 (2021).
>
> [3] Anonymous ICLR 2022 submission, “Leveraging Redundancy in Attention with Reuse Transformers”.
> (https://openreview.net/forum?id=V37YFd_fFgN)

---

### Official Review · Reviewer_kKif · 2021-11-02

**Correctness:** 3
**Technical Novelty And Significance:** 3
**Empirical Novelty And Significance:** 3
**Recommendation:** 8
**Confidence:** 4

**Main Review:**

The paper is very well presented with a large battery of carefully designed experiments and conclusions.
In my opinion, It can naturally be divided into 2 main parts, the study of the MHSA and the layer-wise attention map reuse.

The first part, namely, the analysis of the different emerging patterns of the multi-head self-attention on each layer is very well supported and very interesting. For instance, the layer-wise analysis of phonetic attention was carefully designed to avoid the diagonal contributions of the same phoneme, and the CAD was proposed to account for the cumulative diagonally of layers. The authors found and provide supporting evidence of 2 different regimes in the MHSA. In the upper layer they show that there is a strong diagonal attention which they attribute to linguistic localisation. The evidence for such an intuitive hypothesis is vague in this specific work, and leverages conclusions from other papers, however, they don't provide a strong experimental evidence for that claim. In practice, this is just a semantic issue as there is strong evidence for the emergence of such diagonal pattern in the paper and previous works.

 The lower layers that show a lesser diagonal behaviour are studied from the phonetic perspective, which this is typically ignored. They then hypothesise that those lower layers are performing phonetic localisation and provide evidence for that by:
 1 - adding probes of clarifiers after each conformer layer and showing that there is a peak of classifier performance for phoneme classification at the middle layer.
 2 - computing the phoneme attention relationship (PAR) per attention heads. I found figure 3 confusing, as there are many white dots --maybe without enough attention probability ?-- not commented in the attention pattern. In contrast, I found Fig 5,  Fig 11 or Fig 12 much more illustrative and clear. In addition,  Figure 5 is an average and it would be interesting to contrast the phonetic properties of each head to understand whether there are emerging patterns. It is not very clear whether the selected output vocabulary is affecting in any way to these patterns as they are using 128 sub-word tokens. Would this pattern hold for 512 or more sub-words tokens ?
3 - computing the PAR sharing the attention maps.

In the second part of the paper, the authors propose to reuse attention maps in these 2 regimes, and they prove essentially that those maps can be reuse while keeping the number of parameters the same. This yields small WER improvements and large speed-ups.
I found the constraint on the number of parameters necessary but I would also have liked to see which is the performance when the maps are shared but the hidden value dimension is not increased, as that would have provided a very interesting point for the prior analysis of the first part of the paper. In addition and while the selected pattern make sense for optimising the search space, looking at the results of the phoneme classifications probes, one would have expected maybe additional non-symmetric share patterns for lower layers, like 1(H4)+1(H4)+6(H4) for instance to help to build the phonetic attention from the convolutional front-end.

Finally, the architecture used in the paper is a Conformer architecture which includes a convolutional layers in between MHSA layers.
Although the authors discuss this potential issue in the appendix, I think it is valuable to discuss it in the main text (maybe I missed it).
Having performed one of the experiments turning off the convolutions at least for the phonetic layers would have been really interesting and would have provided a stronger conclusion.

I really enjoyed the paper.
Nice Job.



**Summary Of The Paper:**

The paper analyses the multi-head self attention behaviour in CTC based ASR models.
They identify the working regimes, namely, phonetic and linguistic localisation.
The paper argues that lower MHSA layers build phonetic attentions patterns while upper layers do that linguistically showing some diagonal pattern similarly to NLP tasks. Based on these findings the authors propose to share attention maps across layers.
This yields some modest improvements in terms of WER but obtains larger speeds-up with respect to a basic Conformer model.
During the study they introduce new measurements such as cumulative attention diagonality (CAD) and phoneme attention relationship (PAR).

**Summary Of The Review:**

The paper is well-written and provides a lot of insights into how CTC models with MHSA learn.
I think this is a very interesting experimental analysis of the properties of the Conformer/Transformer models that can help to understand non-autoregressive end2end ASR models. The claims are very well supported and the practical conclusions and implications are very interesting. The code is provided and it seems possible to reproduce the paper.
I would have liked to see more experiments on the paper but just because the results are very interesting.

---

> ### Author Response · Authors · 2021-11-16
> **Response to Reviewer kKif**
>
> We appreciate the reviewer’s insightful comments. Responses to your questions are provided below.
>
> **Response to Major questions:**
>
> (1)	No strong experimental evidence that diagonal attention on upper layers attributes to linguistic localization.
>
> As you mentioned, previous works on natural language processing have shown strong empirical evidence for the diagonal attention pattern, especially in upper layers [1][2]. In contrast, we focused on the lower layers that have low diagonality and discovered that the attention pattern can be explained with phonetic localization, which seemed very critical in performance improvement of Transformer-based ASR.
>
> (2)	Would these patterns hold for 512 or more sub-word tokens?
>
> We choose 128 as the number of tokens following previous works [3][4][5] that employed the sub-word vocabulary size of 128 or 256 for CTC-based LibriSpeech experiments. We also found that the baseline with 128 tokens performed best in our initial experiments. We expect the attention map reuse for phonetic localization might not be affected by the token size change, although the linguistic localization might be affected more than the phonetic localization.
>
> (3)	Performance when preserving the hidden V dimension.
>
> Thank you for your interesting suggestion. In our earlier experiments, we have tested the prototype that preserves the hidden dimension of V. However, the variant, with much fewer parameters, did not show satisfying performance. We believe the impact of attention map reuse can be maximized when the (potential) representative power of the model, partially supported by the number of the parameters, is preserved.
>
> (4)	Possibly additional non-symmetric patterns (1+1+6) for lower layers.
>
> We appreciate your intuitive suggestion. Our experiments in Table 2 mainly focus on evaluating the effect of attention reuse on lower and upper layers with different configurations. We agree that searching for the optimal reuse configuration is practically an important problem to explore.
>
> (5)	Turning off the convolutions (in Conformer) at least for the phonetic layers would have been interesting.
>
> We choose the Conformer model because it shows state-of-the-art performance on ASR. We believe the practical impact of our paper can be maximized when applied to the best performing model. Inspired by your suggestion, we plan to evaluate the model variants (no convolution, or smaller kernel size) and see if the proposed attention reuse is also effective.
>
> **Response to Minor questions:**
>
> (1)	Uncommented white dots in Figure 3?
>
> Thank you for your suggestion. As you noticed, the white points in Fig. 3 correspond to the high probability relationship between frames. To provide an intuitive concept of phonetic localization, we only draw boxes on some representative patterns observed in Fig 3. We added this explanation to the caption.
>
> (2)	Maybe move the discussion of the Conformer to the main text? Figures 5, 11, or 12 are more illustrative and clearer.
>
> We appreciate your suggestion. We will try to reallocate information within the page limit.
>
> *References:*
>
> [1] Guan, Yue, et al. "How Far Does BERT Look At: Distance-based Clustering and Analysis of BERT’s Attention." Proceedings of the 28th International Conference on Computational Linguistics. 2020.
>
> [2] Kovaleva, Olga, et al. "Revealing the Dark Secrets of BERT." Proceedings of the 2019 Conference on Empirical Methods in Natural Language Processing and the 9th International Joint Conference on Natural Language Processing (EMNLP-IJCNLP). 2019.
>
> [3] Majumdar, Somshubra, et al. "Citrinet: Closing the Gap between Non-Autoregressive and Autoregressive End-to-End Models for Automatic Speech Recognition." arXiv preprint arXiv:2104.01721 (2021).
>
> [4] Burchi, Maxime, and Valentin Vielzeuf. "Efficient conformer: Progressive downsampling and grouped attention for automatic speech recognition." arXiv preprint arXiv:2109.01163 (2021).
>
> [5] Kuchaiev, Oleksii, et al. "Nemo: a toolkit for building ai applications using neural modules." arXiv preprint arXiv:1909.09577 (2019).

---

### Official Review · Reviewer_ki2s · 2021-11-03

**Correctness:** 3
**Technical Novelty And Significance:** 3
**Empirical Novelty And Significance:** 2
**Recommendation:** 6
**Confidence:** 4

**Main Review:**

I thank the authors for the additional results and clarifications. Overall, I still feel that the analysis is interesting, but the experimental results are somewhat lacking. It will be interesting to see whether the analysis holds for more popular end-to-end models.

---------------------

Strengths:
* The authors present an excellent analysis of how attention maps are distributed across layers, and find interesting patterns – that lower layers present phonetic information and upper layers more local linguistic information. The figures present enough support for the claim.
* The authors are able to reduce the inference computation by a factor of 1.96 for a 30 second audio, with marginal improvements in WER.

Weaknesses:
* Increasing diagonality of the attention maps has been observed in prior work as well (as the authors have cited and discussed). While the present work improves this understanding (mainly the phonetic information of the lower layers part), the analysis is not entirely novel.
* There are several approaches to improving inference speed by directly addressing the attention map computation. For example, for ASR, prior work has looked at using masked attention where-in each frame only attends to L frames to left and R frames to the right [1, 2, 3]. Therefore, each frame is computing attention only using at (L + 1+ R) frames. This addresses the quadratic increase in computation.
* Alternatively, Performers [4] also proposes linear time and space complexity for attention computation. It would be useful to have some of these comparisons to the presented method.
* The authors’ analysis suggests that the attention map in lower layers is reusable, but those in the upper layers is not (Fig. 5(b)). But their best strategy ties maps for both lower and upper layers. This feels counterintuitive. Is there a reason why that is so?
* How much of the analysis is tied to the CTC-ASR model used? What about LAS [5] or RNN-T [6] models, which are widely used as well?
* The results on Librispeech are much worse than other state-of-the-art (SOTA) end-to-end models (e.g., [7]), most likely because the authors use CTC. It would be useful to see if the techniques result in SOTA performance when using LAS or RNN-T.

Other remarks:
* Figure 1 has too much information and is a little hard to follow. Consider either improving the figure description, or splitting into multiple parts.
* Sec. 3.2: The diagonality analysis is not that different from Zhang et al., and arrives at very similar conclusions. For the models authors consider, are the plots different when using the metric in Zhang et al.?
* Sec. 3.2: Furthermore, it’s not very clear from the description what r(T - 1) is.
* For the plots, is PAR and CAD summarized per utterance or per corpora?
* Figure 6 and the description is hard to follow. What does it mean to set averaged PAR of the baseline to 1.0? Consider moving some of the details from the Appendix to the Sec. 3.4 since PAR is one of the main proposals of the paper.
* Figure 6: The authors claim that a coverage of 0.974 is bad. It’s unclear why. 0.974 seems quite high.
* Does adding more heads to the baseline improve performance? It seems like using 8 heads works the best for the shared attention map model, but the baseline only uses 4.

Minor:
* Abstract: “We propose a novel metric …” The sentence is too long and a little hard to follow, especially with so little context since it is appearing in the abstract.
* Abstract: standardizes the phonetic variance -> reduces the phonetic variance?
* Introduction: 30 seconds … 750 frames: Will be worth adding what is the processing window size (40 msec?).
* Sec. 2.1: (d_h)^2: Consider using special chars like \dag or \ddag for footnotes to avoid confusion. As of now, it looks like d_h is squared.
* Sec. 3.2: What does h stand for in CAD_h?
* Sec 3.3: (1) First, … (2) Second, .. : First and Second are redundant.
* Consider using just 1 decimal point for results. 2 decimal points does not really add a lot of value, comparison-wise.


References:

[1] Zhang, Qian, Han Lu, Hasim Sak, Anshuman Tripathi, Erik McDermott, Stephen Koo, and Shankar Kumar. "Transformer transducer: A streamable speech recognition model with transformer encoders and rnn-t loss." In ICASSP 2020-2020 IEEE International Conference on Acoustics, Speech and Signal Processing (ICASSP), pp. 7829-7833. IEEE, 2020.

[2] Audhkhasi, Kartik, Tongzhou Chen, Bhuvana Ramabhadran, and Pedro J. Moreno. "Mixture Model Attention: Flexible Streaming and Non-Streaming Automatic Speech Recognition." Proc. Interspeech 2021 (2021): 1812-1816.

[3] Tripathi, Anshuman, Jaeyoung Kim, Qian Zhang, Han Lu, and Hasim Sak. "Transformer transducer: One model unifying streaming and non-streaming speech recognition." arXiv preprint arXiv:2010.03192 (2020).

[4] Choromanski, Krzysztof, Valerii Likhosherstov, David Dohan, Xingyou Song, Andreea Gane, Tamas Sarlos, Peter Hawkins et al. "Rethinking attention with performers." arXiv preprint arXiv:2009.14794 (2020).

[5] Chan, William, Navdeep Jaitly, Quoc Le, and Oriol Vinyals. "Listen, attend and spell: A neural network for large vocabulary conversational speech recognition." In 2016 IEEE International Conference on Acoustics, Speech and Signal Processing (ICASSP), pp. 4960-4964. IEEE, 2016.

[6] Graves, Alex. "Sequence transduction with recurrent neural networks." arXiv preprint arXiv:1211.3711 (2012).

[7] Gulati, Anmol, James Qin, Chung-Cheng Chiu, Niki Parmar, Yu Zhang, Jiahui Yu, Wei Han et al. "Conformer: Convolution-augmented transformer for speech recognition." arXiv preprint arXiv:2005.08100 (2020).


**Summary Of The Paper:**

The paper explores how self attention maps in conformer based ASR models are distributed across layers. The authors note that the lower layers of the model capture phonetic information, and the variance of the attention map distribution for the same class of phones across the utterance goes down. The higher layers do not display this behavior. The authors hypothesize that the higher layers capture linguistic information that likely combines information across phones.

Based on the observation that the variance of the attention map distribution goes down, the authors propose reusing the attention map from one layer across multiple layers. This reduces the amount of computation needed for inference, and reduces the training time. Tying attention maps also results in small improvements in performance.


**Summary Of The Review:**

Overall, the authors present an interesting analysis of the attention maps, which in the reviewer’s opinion is the strongest part of the work. It is interesting to categorize lower layers as phonetic, and upper layers as linguistic. That being said, similar analysis has been done in prior work. The presented results are also lacking; it is unclear if the method is very specific to CTC and whether it generalizes to other more popular end-to-end techniques. The authors have also not considered alternative strategies to minimize computation that directly addresses the quadratic computation of attention on long sequences.

---

> ### Author Response · Authors · 2021-11-16
> **Response to Reviewer ki2s**
>
> We appreciate your valuable feedback. Responses to your questions are provided below.
>
> **Response to Major questions:**
>
> (1)	(In summary part) The variance of the attention map distribution for the same class of phones goes down. Attention map in lower layers is reusable, but those in the upper layers are not.
>
> We showed that the attention acts differently in the lower and upper layers. For the upper layers, their attention maps are very similar as their patterns are highly diagonal (Fig. 2). Therefore, the attention maps of upper layers are easily reusable. For the lower layers, reusing the maps is possible not because their attention patterns are lookalike, but their roles can be merged. For example, one attention head may focus on a phoneme “S” while the other may focus on “A”. (Fig. 12) When we reuse the attention map for lower layers, we found that the role of multiple heads can be integrated into fewer heads (ex: catching both “S” and “A”). The PAR coverage (Fig. 6) evaluates how much the phonetic relationship can be recovered when attention map is reused. If there are too few unique heads left or the capability of each head is insufficient, the performance cannot be preserved, as shown in the configuration of 8x2.
>
> (2)	Increasing diagonality of the attention maps has been observed in prior works. What is the difference between the diagonality analysis in Zhang et al. [1]?
>
> We first emphasize that our main finding is about the phonetic localization in the lower layers, where the diagonality does not linearly increase. Our CAD metric dearly captures the flat region in lower layers (Fig. 2). In contrast, in [1], the diagonality increases from the lower layers to the upper layers, which may not be sufficient observation in ASR.
>
> As a metric, the CAD is more comprehensive because it provides the overall diagonality value as well as the tendency of the attention according to the distance (Fig. 9). For example, one can use the information to determine the attention restriction range (L/R frames). We discussed the difference in Appendix B.1.2 in the initial version, and we added the above discussion in the revised version.
>
> (3)	Is the PAR coverage of 0.974 bad? What does it mean to set averaged PAR of the baseline to 1.0?
>
> We empirically found that even a small drop in PAR correlates with considerable accuracy degradation. We found that successful attention reuse always achieved PAR coverage greater than 0.99. In Fig. 13, we visualized that the coverage ratio over 0.99 (2x8, 4x4) successfully recovered the baseline PAR but the value of 0.974 (8x2) failed to learn some important relationships, which caused the noticeable performance loss.
>
> (4)	Other approaches that directly address the attention map computation for inference speed. (Masked attention, Performers, etc.)
>
> Thank you for pointing out the important references. As you mentioned, there have been various approaches to reduce the heavy computational burden of self-attention. But this work is one of the first to connect the computation saving methods (=attention map reuse) and the mechanism behind it. We believe that understanding the pattern of attention maps can be connected to the different ways of attention reuse (ex: tradeoff between the number of heads versus inference speed).
>
> (5)	Is this analysis also applied to LAS or RNN-T?
>
> Thank you for your suggestion. We are planning to apply attention map reuse for LAS/RNN-T as future works. Because LAS and RNN-T have additional modules such as decoder or prediction network, the encoder (acoustic model) may not behave in the exact same way. However, we strongly expect that the phonetic localization at the lower layers would also appear in LAS/RNN-T, and attention reuse could be beneficial.
>
> (6)	Does adding more heads (H=8) to the baseline improve performance?
>
> Our baseline model follows the default setting (H=4) from the Conformer paper. It would be an interesting question whether the performance of baseline can be improved by varying the number of heads in the Conformer-CTC setting. However, we expect the attention would behave the same for the improved baseline, therefore it can also benefit from the attention map reuse.
>
> **Response to Minor points:**
>
> (1)	Suggestions on figures, abstract, introduction, decimal points, footnote, etc.
>
> Thank you for your kind advice. We applied the changes in the revised version.
>
> (2)	Is PAR and CAD summarized per utterance or per corpora? r(T-1) is not very clearly explained.
>
> PAR and CAD are averaged through the corpora. We denoted this in the revised version. For the term r(T-1), please see our response to (Q1, Reviewer Vyqx) for a detailed explanation.
>
> *References:*
>
> [1] Zhang, Shucong, et al. "On the usefulness of self-attention for automatic speech recognition with transformers." 2021 IEEE Spoken Language Technology Workshop (SLT). IEEE, 2021.

---

### Official Review · Reviewer_Vyqx · 2021-11-03

**Correctness:** 4
**Technical Novelty And Significance:** 4
**Empirical Novelty And Significance:** 4
**Recommendation:** 8
**Confidence:** 4

**Main Review:**

# Correctness and Significance

The paper is logical and methodical in the investigation. The proposed measurements and experiments make sense. The CAD metric makes sense. This won't change the result of the work much, but CAD looks unnormalized: the double sum seems to require 1/T^2 term, not 1/T.

The phoneme analysis is conducted well. Both the phoneme classification accuracy curve and the phoneme attention relationship metrics confirm the paper's hypothesis on the phonetic-linguistic division. Nevertheless, I would like to see a discussion on alternative hypotheses. I don't have a concrete suggestion here, but is it possible that the result obtained in Figure 2 is spurious or can be explained by something else.

Finally, the proposed architecture for the attention map reuse is a logical step to improve the model. The paper reports multiple architecture variations and compares them. I found the most inspiring that this technique provides a way to reduce latency.

# Clarity

The paper is very well structured and well written. It was a pleasure to read it.

# Minor things

- The graphs are hard read when printed. Especially in black and white.
- Session -> Section

**Summary Of The Paper:**

The paper consists of two parts. First, it conducts the analysis of an ASR model. Specifically, it measures the diagonality of the self-attention map and notes that the diagonality for the top layers is higher than for the lower layers. Based on this, the paper suggests that the lower layers are responsible for the phonetic information and the higher layers are responsible for the linguistic information. Then, the paper confirms this hypothesis by using each layer for a phoneme classifier.

Second, based on the above observations, the paper proposes a method to exploit this division into the phonetic-linguistic layers. The proposed method is to reuse the attention maps across several layers. This modification allows for more efficient architecture both in terms of latency in train and test time. In some cases, the proposed architecture outperforms the baseline.

**Summary Of The Review:**

In summary, this is a good exploration into the transformer architecture for speech recognition. The paper provides valuable insights into the transformer's attention map structure and connects it with the phonetics. Then the paper uses these insights to improve the model.

I would like to request the alternative hypotheses on why the attention becomes diagonal in higher layers. Or some discussion on this.

Then, the figures need to be improved for reading when printed.

---

> ### Author Response · Authors · 2021-11-16
> **Response to Reviewer Vyqx**
>
> We appreciate the reviewer’s valuable feedback. Responses to your comments are provided below.
>
> **Response to Major points:**
>
> (1) The CAD metric looks unnormalized.
>
> Thank you for carefully checking the details. The CAD is a good indicator of how fast the accumulated attention increases over the distance, directly represents the diagonality of the attention weight. The CAD metric can be interpreted as the area under the function $D(r)$, where $D(r)$ calculates the amount of total attention weight within the restricted range $j \in [i-r(T-1),i+r(T-1)]$. If the $r$ is same, a larger $D(r)$ means that the attention is more concentrated near the diagonal. Please note that $D(r)$ is a monotonically increasing function whose output is always in the range $[0, 1]$.
>
> Specifically, considering the attention weight A, the sum(j) accumulates the attention weights of each row, and 1/T * sum(i) averages this value through multiple rows (Eq. 3). Since CAD computes the area under the cumulative attention, we consider that the second 1/T normalization is not necessary. We added additional explanations above in Appendix B.1.1.
>
> (2) Alternative hypothesis on why the attention becomes diagonal in higher layers.
>
> The output unit of ASR task is a sub-word token that consists of multiple phonemes. Therefore, it is natural to mainly attend to local phonetic context to produce sub-word outputs, after sufficient phonetic localization is achieved in lower layers. A recent study on the state-less RNNT [1] implicates that the long-range context near the output of the network may not be necessary when using sub-word as output unit, while the work is not directly applicable to ours.
>
> We also suggest that the local context is very important in learning linguistic characteristics. For example, previous analyses in NLP domain, such as BERT or NMT, also reported a considerable number of diagonal attention patterns to exist in self-attention layers [2][3][4].
>
> **Response to Minor points:**
>
> Thank you for your suggestions. We improved the graphs to be more distinguishable in black & white and fixed the mentioned typos.
>
> *References:*
>
> [1] Ghodsi, Mohammadreza, et al. "RNN-transducer with stateless prediction network." ICASSP 2020-2020 IEEE International Conference on Acoustics, Speech and Signal Processing (ICASSP). IEEE, 2020.
>
> [2] Park, Cheonbok, et al. "Sanvis: Visual analytics for understanding self-attention networks." 2019 IEEE Visualization Conference (VIS). IEEE, 2019.
>
> [3] Gong, Linyuan, et al. "Efficient training of bert by progressively stacking." International Conference on Machine Learning. PMLR, 2019.
>
> [4] Kovaleva, Olga, et al. "Revealing the Dark Secrets of BERT." Proceedings of the 2019 Conference on Empirical Methods in Natural Language Processing and the 9th International Joint Conference on Natural Language Processing (EMNLP-IJCNLP). 2019.

---

### Comment · Area_Chair_Esss · 2021-11-15
**Please address the reviewers' comments**

Hi Authors,

Please address the reviewers' comments. Thanks!

---

### Author Response · Authors · 2021-11-21
**Additional experimental results**

We really appreciate the invaluable comments, feedback, and suggestions from the reviewers. Inspired by the discussion, we conduct additional experiments to address possible concerns.

We train the models with the same setting as the Conformer-M baseline (as in the paper) but with a limited attention range for either lower or upper layers. Among the 16 layers in the baseline, lower and upper layers consist of 8 layers each and correspond to phonetic localization and linguistic localization, respectively. For selected layers, we restrict each frame to only attend to near neighbors within the distance of 64 frames. These models (Low64, Up64) are trained from scratch. We compare the WER on the LibriSpeech dataset below:

| Model    | Lower layers | Upper layers | dev-clean | dev-other | test-clean | test-other |
|----------|--------------|--------------|-----------|-----------|------------|------------|
| Baseline | unlimited    | unlimited    | 3.1       | 8.3       | 3.2        | 8.4        |
| Low64    | [-64, 64]    | unlimited    | 3.1       | 8.4       | 3.4        | 8.5        |
| Up64     | unlimited    | [-64, 64]     | 3.1       | 8.2       | 3.3        | 8.2        |
| 4(H8)x4  | unlimited    | unlimited    | 3.1       | 8.1       | 3.2        | 8.1        |

For lower layers, attention range restriction causes a clear degradation of the performance. We consider that phonetic localization in lower layers demands a wide range of attention, and is critical for lowering WER. On the other hand, for upper layers, attention range restriction shows almost comparable performance to the baseline, and even better in some subsets. We hypothesize that the upper layers for CTC-based ASR may not require a very wide context, because their attention pattern is highly diagonal. To the best of our knowledge, we first conduct attention range restriction differently for lower and upper layers. We included the results in the (second) revised version of our paper.

**For reviewer Vyqx**

We hope this experiment provides some empirical evidence for your comment on the diagonality of the upper layers. As the models are trained from scratch, we believe the result above supports that the long-range context may not be essential for linguistic localization, at least for CTC-based ASR models.


**For reviewer ki2s**

As you mentioned the masked attention methods as an alternative approach to save the computation, we hope the results can be a good comparison. We show that the restriction (masked attention) methods could minimize the impact on the performance by considering the role of self-attention for each layer.

In addition, the restriction-based computational savings seems to be no larger than the proposed attention reuse. A context range of 128 (64+64) frames corresponds to about 5.1 seconds, where the average utterance length of the corpora is about 7.4 seconds. Therefore, we expect approximately 30% reduction in attention calculation in case the masked attention is applied to every layer. Please consider that our attention reuse (4x4) reduces about 75% of the computation for attention without causing any degradation in the recognition accuracy.

**For reviewer kKif**

We hope this experiment can be good empirical evidence that the linguistic localization for CTC-based ASR is sufficient with a relatively short range of attention, for example, diagonal pattern. Although this experiment does not fully explain the emergence of such diagonal patterns in upper layers, we suggest this can be a valuable observation.

**For reviewer mbFx**

The comparison between Low64 and Up64 cases demonstrates that the ASR model design can benefit from our findings on phonetic and linguistic localization. In this case, we can take different restriction range for lower and upper layers, and the analysis of CAD can help decide the proper value. Without the knowledge, it would take much more trial and errors to design the best-working tradeoff.

---

> ### Comment · Reviewer_ki2s · 2021-11-29
> **computational reduction**
>
> Can you clarify how the proposed 4x4 architecture saves 75% of the computation? From Tab. 3, it looks like for a 10 sec audio, latency reduction is ~30%. Are you referring to the #ops? In terms of latency, how much latency savings does masked attention provide for a 30 second audio?

---

> > ### Author Response · Authors · 2021-11-29
> > **Response to the computational reduction**
> >
> > Dear reviewer, we are sorry for not clearly denoting the meaning of values. The aforementioned 30% and 75% reduction are the approximation of ops reduction for dot-product in self-attention (SA) calculation. The computational burden of SA is proportional to TxTxN (T is the sequence length, N is the number of layers). For masked attention, this cost becomes (L+R+1)xTxN, and for attention reuse 4x4, the cost is TxTx(N/4).
> >
> > The values in Table 3 indicate the actual latency evaluated on GPU, including projections, scaled dot-products, and feed-forward modules. Therefore, the speedup on GPU would be smaller than only considering the dot-product, for both masked attention and attention reuse.
> >
> > To fully benefit from the restricted range of masked attention, we need a specialized kernel that efficiently reorders and only computes (L+R+1) values for each frame to perform dot-product as dense matrix multiplication. For example, although not exactly the same, BigBird[1] optimizes the GPU kernel for the block-wise window local attention (see Figure 4(c) of [1]). We only compared the ops reduction because the specific kernel for Conformer encoder is not prepared. We expect the additional overhead of gather & scatter, or masking, without such kernel.
> >
> > [1] Zaheer, Manzil, et al. "Big Bird: Transformers for Longer Sequences." NeurIPS. 2020.

---

> ### Comment · Reviewer_kKif · 2021-11-29
> **Interesting added experiments**
>
> Dear Authors,
> thank you very much for launching additional experiments.
> The new experiments clearly add value to the discussion and paper and add another value empirical evidence supporting the paper hypotheses.

---

### Comment · Area_Chair_Esss · 2021-11-24
**Please update your ratings if needed based on the authors' responses**

Dear Reviewers,

The authors have made detailed responses to all the reviews. Please take a look and see whether they address your concerns and update the ratings if necessary. Thanks for your help and expertise!

---

### Decision · Program_Chairs · 2022-01-20

**Decision:**

Accept (Spotlight)

**Comment:**

The paper conducted a thorough experimental analysis of the attention map in the Conformer models for CTC based speech recognition models and connected it with phonetic and linguistic information in the speech. Using these insights, the paper presented some computation improvement and marginal quality gains. The authors actively conducted additional experiments to further justify the claims. The paper is strong in terms of the systematic way of in-depth analysis and further development (i.e. sharing the attention map across layers for speedup). But as pointed out by the reviewers, it lacks some comparisons with other alternatives to justify the importance of sharing attention maps in reducing computations.  Also it would be better if there's justifications on how the observations generalize to other types of models (such as LAS, RNN-T).

The decision is mainly because of the thorough analysis conducted in the paper which can be a good contribution to the community.